# Habitual Miso (Fermented Soybean Paste) Consumption Is Associated with Glycemic Variability in Patients with Type 2 Diabetes: A Cross-Sectional Study

**DOI:** 10.3390/nu13051488

**Published:** 2021-04-28

**Authors:** Fuyuko Takahashi, Yoshitaka Hashimoto, Ayumi Kaji, Ryosuke Sakai, Akane Miki, Takuro Okamura, Noriyuki Kitagawa, Hiroshi Okada, Naoko Nakanishi, Saori Majima, Takafumi Senmaru, Emi Ushigome, Masahide Hamaguchi, Mai Asano, Masahiro Yamazaki, Michiaki Fukui

**Affiliations:** 1Department of Endocrinology and Metabolism, Graduate School of Medical Science, Kyoto Prefectural University of Medicine, 465, Kajii-cho, Kawaramachi-Hirokoji, Kamigyo-ku, Kyoto 602-8566, Japan; fuyuko-t@koto.kpu-m.ac.jp (F.T.); kaji-a@koto.kpu-m.ac.jp (A.K.); sakaryo@koto.kpu-m.ac.jp (R.S.); aknmk623@koto.kpu-m.ac.jp (A.M.); d04sm012@koto.kpu-m.ac.jp (T.O.); nori-kgw@koto.kpu-m.ac.jp (N.K.); conti@koto.kpu-m.ac.jp (H.O.); naoko-n@koto.kpu-m.ac.jp (N.N.); saori-m@koto.kpu-m.ac.jp (S.M.); semmarut@koto.kpu-m.ac.jp (T.S.); emis@koto.kpu-m.ac.jp (E.U.); mhama@koto.kpu-m.ac.jp (M.H.); maias@koto.kpu-m.ac.jp (M.A.); masahiro@koto.kpu-m.ac.jp (M.Y.); michiaki@koto.kpu-m.ac.jp (M.F.); 2Department of Diabetology, Kameoka Municipal Hospital, 1-1 Noda, Shinochoshino, Kameoka 621-8585, Japan; 3Department of Diabetes and Endocrinology, Matsushita Memorial Hospital, 5-55 Sotojima-cho, Moriguchi 570-8540, Japan

**Keywords:** fermented soy foods, glycemic control, glycemic variability, miso, type 2 diabetes

## Abstract

Glycemic control, including glycemic variability, is important for the prevention of diabetic vascular complications in patients with type 2 diabetes mellitus (T2DM). There was an association between miso soup intake and insulin resistance. However, the relationship between habitual miso consumption and glycemic control, including glycemic variability, in patients with T2DM remains unknown. We defined people without habitual miso consumption if they did not consume miso soup at all in a day. The average, standard deviation (SD), and coefficient of variation (CV), calculated as CV = (SD/average HbA1c) × 100 (%), of hemoglobin (Hb) A1c levels were evaluated. The proportions of habitual miso consumption of male and female were 88.1% and 82.3%, respectively. The average (7.0 [6.4–7.5] vs. 7.3 [6.8–8.4] %, *p* = 0.009), SD (0.21 [0.12–0.32] vs. 0.37 [0.20–0.72], *p* = 0.004), and CV (0.03 [0.02–0.04] vs. 0.05 [0.03–0.09], *p* = 0.005) of HbA1c levels in female with habitual miso consumption were lower than those of female without. Moreover, habitual miso consumption correlated with average (β = −0.251, *p* = 0.009), SD (β = −0.175, *p* = 0.016), and CV (β = −0.185, *p* = 0.022) of HbA1c levels after adjusting for covariates. However, no association between habitual miso consumption and any glycemic parameters was shown among male. This study clarified the association between habitual miso consumption and good glycemic control, including glycemic variability, in female, but not in male.

## 1. Introduction

Glycemic control is important for the prevention of diabetic vascular complications in patients with type 2 diabetes mellitus (T2DM) [1,2]. Glycemic variability is important for preventing these complications, independent of the average glycemic control [3,4]. Hemoglobin (Hb) A1c variability suggests insulin resistance and the presence of visceral adipose tissue [5]. A recent meta-analysis revealed that HbA1c variability was associated with renal disease, cardiovascular disease (CVD), and all-cause mortality in patients with T2DM and retinopathy [6]. Therefore, we need to focus not only on the mean glycemic control, but also on glycemic variability, especially HbA1c variability, as treatment targets in patients with T2DM.

Miso, a traditional Japanese food made by fermenting soybeans, is very popular in Japan. It contains not only vegetable proteins, carbohydrates, and fats, but also minerals, vitamins, and microorganisms [7]. A previous review revealed that miso can inhibit the incidence of cancers in mice and rats [8]. Other studies have also shown the protective effect of miso on the incident of hypertension in Japanese people without hypertension [9,10]. Moreover, it has been reported that there is an association between habitual miso soup consumption and lower insulin resistance [11]. However, the relationship between habitual miso consumption and glycemic control in patients with T2DM remains unknown. Furthermore, a relationship between habitual miso consumption and glycemic variability in patients with T2DM has not been previously reported. Therefore, in this cross-sectional study of patients with T2DM, we researched the impact of habitual miso consumption on blood glucose control, including glycemic variability, in male and female.

## 2. Method

### 2.1. Study Participants

We commenced a prospective cohort study (KAMOGAWA-DM cohort study) in 2014, and it is currently ongoing [12]. Outpatients of the Kyoto Prefectural University of Medicine Hospital (Kyoto, Japan) and Kameoka Municipal Hospital (Kameoka, Japan) for diabetes was included in this cohort study. All included patients signed informed consent. This study was approved by the local research ethics committee (No. RBMR-E-466-6) and performed according to the Declaration of Helsinki. The inclusion criterion of this cross-sectional study was the ability to answer the questionnaires completely, including brief-type self-administered diet history questionnaire (BDHQ), from January 2016 to December 2018. The exclusion criteria were as follows: patients without T2DM, those with incomplete or inadequate responses to the questionnaire, and patients whose HbA1c level was measured fewer than five times per year [13].

### 2.2. Data Collection

Family history of diabetes, exercise habits, smoking status, and duration of diabetes were assessed using a uniform questionnaire. Patients were categorized as smokers or non-smokers. Patients who regularly exercised were defined as those who regularly played any kind of sport >1×/week.

Body mass index (BMI) and ideal body weight (IBW) [14] were calculated as weight (kg) divided by height squared (m)^2^ and 22 × (height [m])^2^.

Venous blood was gathered from people with overnight fasting, and the levels of fasting blood sugar, triglycerides, high-density lipoprotein (HDL) cholesterol, uric acid, and creatinine (Cr) were measured. Triglycerides to HDL cholesterol ratio was calculated as triglycerides (mmol/L) divided by HDL cholesterol (mmol/L) [15]. Estimated glomerular filtration rate (eGFR) was calculated: eGFR (mL/min/1.73 m^2^) = 194 × Cr^−1.094^ × age^−0.287^ (×0.739, if woman) [16]. HbA1c levels were estimated using high-performance liquid chromatography and expressed as NGSP units. Blood pressure was automatically measured using a HEM-906 device (OMRON, Kyoto, Japan) after the participants rested for 5 min in a quiet space. Furthermore, data on medications, including those for usage of insulin and antihypertensives, were collected from the patients’ medical records. Patients with hypertension were defined as those with systolic/diastolic blood pressure ≥140/90 mmHg and/or those using antihypertensive drugs.

### 2.3. Definition of Glycemic Parameters

We measured HbA1c at every visit. The HbA1c levels, measured throughout the year, were obtained from electronic medical records. The average level, the standard deviation (SD), and the coefficient of variation (CV), defined as (SD/average HbA1c) × 100 (%), of HbA1c were then assessed. 

### 2.4. Data of Habitual Intake of Diet Intake

A BDHQ was used to evaluate habitual food and nutrient intake in the previous month [17]. The specifications and validity of BDHQ were reported previously [17]. Data of the intake of energy, carbohydrates, proteins, fat, and fiber as well as miso soup and alcohol consumption were obtained using BDHQ. Frequency of miso soup intake was how often they consume miso soup per day; none, less than 1 cup, 1 cup, 2 cups, 3 cups, 4 cups, 5 cups, 6 or 7 cups, or 8 cups or more per day. We defined people without habitual miso consumption if they did not consume miso soup at all in a day [18]. 

### 2.5. Statistical Analysis

Data are shown as mean (SD), median [interquartile range] when the variables were skewed. A *p*-value of <0.05 was considered to be statistically significant.

Participants were categorized into two groups based on habitual miso consumption. Because the characteristics and dietary intakes were different between sex, the data were analyzed male and female, separately. Evaluation of normal distribution was performed by Shapiro-wilk normality test. Student’s *t*-test or Mann-Whitney U test for continuous variables and chi-square test for categorical variables was used for evaluation of the differences. Evaluation of correlation was assessed by Pearson’s correlation coefficient. Because the miso soup intake was a skewed variable, logarithmic transformation was done, before performing the investigation of the relationship of glycemic variability and log (miso soup intake +1). 

To investigate the effect of habitual miso consumption on glycemic parameters, including average, SD, and CV of HbA1c, multiple regression analyses was used, adjusting for potential cofounders; age, BMI, duration of diabetes, average HbA1c, insulin treatment [19], exercise habit, smoking status, intake of energy [20], intake of carbohydrate, and intake of dietary fiber [21]. Statistical analyses were performed by JMP software (version 13.2; SAS Institute Inc., Cary, NC, USA) and EZR (Saitama Medical Center, Jichi Medical University, Saitama, Japan) [22]. 

## 3. Results

This study included 523 patients (276 male, 247 female). Among them, 121 patients (57 male and 64 female) with no data on BDHQ, 40 patients (20 male and 20 female) without T2DM, and 72 patients (39 male and 53 female) whose HbA1c levels were measured fewer than five times per year were excluded. Therefore, 290 patients (160 male and 130 female) were included in the study (Figure 1).

Table 1 represent the baseline clinical characteristics of the study participants. Median age and BMI were 68.0 [63.0–74.0] years and 23.5 [21.6–25.8] kg/m^2^ in male and 68.0 [62.0–73.8] years and 23.6 [21.3–26.3] kg/m^2^ in female. The average, SD, and CV of HbA1c were 7.0 [6.6–7.5] %, 0.21 [0.14–0.33], and 0.03 [0.02–0.05] in male and 7.0 [6.5–7.6] %, 0.22 [0.13–0.37], and 0.03 [0.02–0.05] in female, respectively.

BDHQ, brief-type self-administered diet history questionnaire; T2DM, type 2 diabetes mellites.

Table 2 shows the results of dietary intake data. The percentage of habitual miso consumption was 88.1% (n = 141/160) and 82.3% (n = 107/130) in male and female, respectively.

Table 3 represents the clinical characteristics of patients based on habitual miso consumption or not. Among females, the average (*p* = 0.009), SD (*p* = 0.004), and CV (*p* = 0.005) of HbA1c were lower in patients with habitual miso consumption than in those without it. Fasting blood sugar in females with habitual miso consumption was lower than in those without it (*p* = 0.034). The percentage of hypertension was lower in females with habitual miso consumption than in females without (*p* = 0.023). However, there was no association between habitual miso consumption and clinical characteristics in males.

Habitual dietary intake based on the presence of habitual miso consumption are shown in Table 4. The intakes of energy (*p* < 0.001), protein (*p* < 0.001), animal protein (*p* = 0.010), vegetable protein (*p* < 0.001), fat (*p* = 0.031), and dietary fiber (*p* < 0.001) in females with habitual miso consumption were higher than those in females without it. In contrast, there was no difference in dietary habits between male with and without habitual miso consumption. 

The correlation between the log (miso soup intake + 1) and average, SD, or CV of HbA1c were investigated. In females, the log (miso soup intake + 1) was correlated with average (*r* = −0.290, *p* < 0.001), SD (*r* = −0.316, *p* < 0.001) and CV (*r* = −0.289, *p* < 0.001) of HbA1c. On the other hand, the log (miso soup intake + 1) was not correlated with average (*r* = −0.079, *p* = 0.322), SD (*r* = −0.034, *p* = 0.667) and CV (*r* = −0.013, *p* = 0.874) of HbA1c in males. 

To examine the association between habitual miso consumption and glycemic parameters, we performed multiple regression analyses (Table 5). In females, habitual miso consumption was related to the average (β = −0.251, *p* = 0.009), SD (β = −0.175, *p* = 0.016), and CV (β = −0.185, *p* = 0.022) of HbA1c levels after adjusting for covariates. However, habitual miso consumption was not associated with the average, SD, and CV of HbA1c levels in males.

## 4. Discussion

In this study, we examined the relationship between habitual miso consumption and glycemic control in patients with T2DM and found that habitual miso consumption was independently related to glycemic control, including glycemic variability, which is reported to be a risk factor for microvascular and macrovascular complications in female with T2DM [3,4,23]. In this study, 88.1% of male and 82.3% of female were categorized as habitual miso consumption, which was higher than the general population as previously reported [11,24]. There is a possibility that people with T2DM might consume miso more usually than general people.

Fermentation is one of the main processes used to produce food from soybeans. During the production of fermented soybean products, bioactive constituents, including isoflavones and peptides, may alter their effectiveness in the treatment of T2DM [25]. The consumption of soybeans and fermented soybeans may be related to the lower prevalence of T2DM in Asians [26]. Various fermented soybean products are consumed in many Asian countries, including Japan, Korea, China, Indonesia, and Vietnam. In Japan, the most popular fermented soybean products are miso, natto, and soy sauce. Miso suppresses the development of various diseases, and habitual miso soup consumption is related to reduced insulin resistance [8,9,10,11]. Nevertheless, the relationship between miso and glycemic control, including glycemic variability, has not been revealed in patients with T2DM.

The relationship between habitual miso consumption and glycemic control may be explained as follows: 

A previous study showed that fat accumulation in fat mass was suppressed by miso intake and clarified the anti-obesity effects of miso in mice model [27]. In fact, the consumption of miso soup is associated with lower insulin resistance in female [11]. Insulin resistance and visceral fat accumulation have been reported to be associated with glycemic control, including HbA1c variability [5]. This suggests that habitual miso consumption is associated with glycemic parameters through reduced fat accumulation and insulin resistance, which may lower HbA1c variability.

Miso, which are rich in soybean-specific proteins, lecithin, isoflavones, saponin, lipids rich in polyunsaturated fatty acids, and vitamin E [7,28,29], is a product of fermented soybeans. Isoflavones may have beneficial effects for improving glucose homeostasis and diabetes [24]. Isoflavones are similar in structure to estradiol and so can combine to the vascular wall’s estrogen receptor [30]. Moreover, soybean proteins and isoflavones have been reported to reduce visceral fat accumulation in animal models [31,32,33]. In addition, we revealed the association between habitual miso consumption and the abdominal obesity in a previous study [18]. Therefore, isoflavones in miso may have inhibited the accumulation of visceral fat and contributed to good glycemic control in female. A previous study revealed that β-conglycinin, which included in miso, was a beneficial food ingredient for prevention of obesity [34]. In addition, β-conglycinin was reported to decrease triglycerides levels [35,36]. Therefore, patients with habitual miso consumption may prevent visceral fat accumulation and improve glycemic control. However, there was no difference between triglycerides level of people with habitual miso consumption and that of people without in this study. This might be because that this study was an observational study; and thus, the physicians prescribed the medication for dyslipidemia, if the triglycerides level were high. Taken together, these findings suggest that females with habitual miso consumption may have better glycemic control.

Contrarily, habitual miso consumption was not related to glycemic control in male; however, the reason underlying this finding is unclear. Possible explanations might be that habitual miso consumption was not associated with diet quality in male. Consistent with our study, previous research showed a relationship of habitual miso soup consumption with insulin resistance in females, but not in males [11]. In addition, it was previously reported that soy intake may be related to a lower risk of T2DM in Japanese female, but not in Japanese male [37].

However, the limitations of this study should be shown. First, the frequency of miso soup intake was evaluated by self-reported data, and there was a possibility that the data was not accurate. Second, we did not assess the association between habitual miso consumption and insulin concentrations and postprandial blood sugar level in this study. We evaluated triglycerides to HDL cholesterol ratio, which is reported as a marker of insulin resistance [15], and found that there was no association between habitual miso consumption and triglycerides to HDL cholesterol ratio in this study (0.8 [0.5–1.6] vs. 0.8 [0.6–1.4], *p* = 0.924 in male 0.7 [0.5–1.2] vs. 0.9 [0.6–1.2], *p* = 0.426 in female). This might be because the effect of taking medication for dyslipidemia. Thus, it was not unclear that female with habitual miso consumption had lower insulin resistance than those without, and the association between habitual miso consumption and postprandial blood sugar level. Third, there is a possibility that different types of miso would affect the results, since several types of miso, such as rice koji miso, barley koji miso, soybean koji miso, and blended miso, were used in Japan. Unfortunately, however, we did not investigate type of miso in this study. Therefore, further research is needed the association between each type of miso glycemic variability. Fourth, because this was a cross-sectional study and the sample size was relatively small, we need further research, such as a study with more participants and a randomized clinical trial. Finally, this study included only Japanese; thus, it is unclear whether the results can be generalized to other ethnic populations.

## 5. Conclusions

This study demonstrated the relationship between habitual miso consumption and glycemic variability in female, but not in male, with T2DM. Miso, one of the famous traditional Japanese food, has been showed that consumption of miso soup may help in the management of patients with T2DM.

## Figures and Tables

**Figure 1 nutrients-13-01488-f001:**
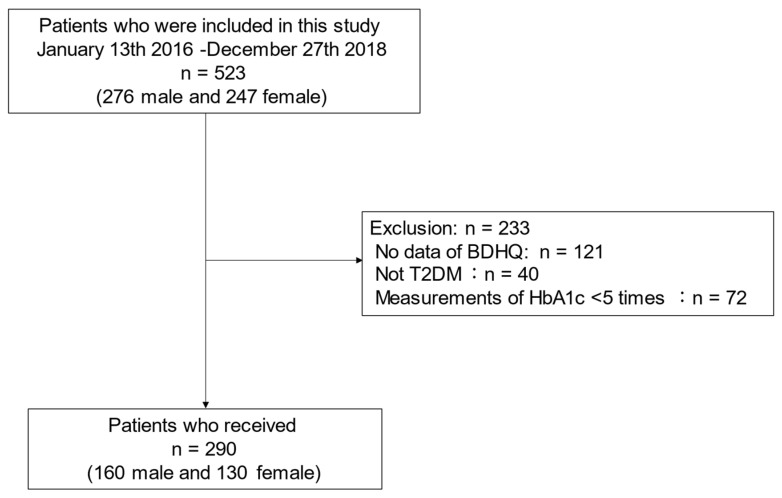
Inclusion and exclusion flow.

**Table 1 nutrients-13-01488-t001:** Clinical characteristic of study patients.

	All *n* = 290	Male *n* = 160	Female *n* = 130	*p*
Age, years	68.0 [62.3–74.0]	68.0 [63.0–74.0]	68.0 [62.0–73.8]	0.491
Diseases duration of diabetes, years	13.0 [7.0–20.0]	14.0 [8.0–21.0]	12.5 [5.3–18.0]	0.110
Family history of diabetes (−/+)	155/135	92/68	63/67	0.157
Height, cm	161.0 (9.2)	167.2 (6.5)	153.3 (5.4)	<0.001
Body weight, kg	61.0 [54.8–68.7]	65.3 [59.2–72.6]	55.7 [51.0–61.8]	<0.001
Body mass index, kg/m^2^	23.5 [21.4–26.0]	23.5 [21.6–25.8]	23.6 [21.3–26.3]	0.481
Systolic BP, mmHg	132.0 [121.0–144.0]	132.0 [122.0–144.0]	132.0 [120.0–143.0]	0.748
Diastolic BP, mmHg	77.5 [71.0–85.0]	80.0 [71.0–86.0]	76.0 [70.0–82.0]	0.055
Antihypertensive drugs (−/+)	123/167	66/94	57/73	0.745
Presence of hypertension (−/+)	94/196	54/106	40/90	0.679
Insulin (−/+)	211/79	118/42	93/37	0.773
Habit of smoking (−/+)	246/44	126/34	120/10	0.002
Habit of exercise (−/+)	140/150	76/84	64/66	0.861
Average HbA1c, mmol/mol	53.3 [48.3–58.9]	53.5 [49.0–58.5]	53.2 [48.0–59.1]	0.544
Average HbA1c, %	7.0 [6.6–7.5]	7.0 [6.6–7.5]	7.0 [6.5–7.6]	0.544
SD of HbA1c	0.21 [0.14–0.34]	0.21 [0.14–0.33]	0.22 [0.13–0.37]	0.899
CV of HbA1c	0.03 [0.02–0.05]	0.03 [0.02–0.05]	0.03 [0.02–0.05]	0.962
Plasma glucose, mmol/L	7.7 [6.5–9.3]	7.9 [6.8–10.0]	7.5 [6.2–8.9]	0.042
Creatinine, umol/L	64.5 [55.0–80.4]	74.7 [63.6–88.4]	55.7 [49.5–63.4]	<0.001
eGFR, mL/min/1.73 m^2^	71.8 [59.0–82.5]	71.6 [57.4–83.6]	72.5 [60.9–81.8]	0.752
Uric acid, mmol/L	297.4 [255.8–350.9]	3188.2 [273.6–364.3]	279.6 [243.9–321.2]	<0.001
Triglycerides, mmol/L	1.2 [0.9–1.9]	1.3 [0.9–1.9]	1.2 [0.9–1.7]	0.698
HDL cholesterol, mmol/L	1.5 [1.3–1.8]	1.4 [1.2–1.7]	1.6 [1.3–1.8]	0.005
Triglycerides/HDL cholesterol, mmol/mmol	0.8 [0.5–1.4]	0.8 [0.5–1.5]	0.8 [0.5–1.2]	0.166

Number (absence [−]/presence [+]), mean (standard deviation) or median [interquartile range] was used for expression of data. Chi-square test, Student’s t test, or Mann-Whitney U test was used for evaluation of the difference between group. CV, coefficient of variation; BP, blood pressure; eGFR, estimated glomerular filtration rate; HDL, high-density lipoprotein; and SD, standard deviation.

**Table 2 nutrients-13-01488-t002:** Habitual diet intake of study participants.

	All *n* = 290	Male *n* = 160	Female *n* = 130	*p*
Intake of total energy, kcal/day	1658.2 [1338.1–2061.3]	1813.4 [1541.5–2180.7]	1483.4 [1188.8–1779.4]	<0.001
Energy intake, kcal/IBW/day	29.2 [23.9–35.8]	30.0 [25.2–36.2]	28.1 [22.0–34.6]	0.137
Intake of total protein, g/day	66.8 [55.5–85.1]	71.5 [57.3–88.2]	62.4 [50.7–82.4]	0.013
Intake of protein, g/IBW/day	1.2 [0.9–1.5]	1.2 [0.9–1.4]	1.2 [1.0–1.6]	0.116
Intake of protein per energy, %	16.2 [14.3–18.9]	15.4 [13.5–17.7]	17.6 [15.4–20.2]	<0.001
Intake of animal protein, g/day	40.8 [30.4–54.0]	41.4 [31.4–56.0]	38.5 [29.5–53.0]	0.152
Intake of animal protein, g/IBW/day	0.7 [0.5–1.0]	0.7 [0.5–0.9]	0.8 [0.5–1.0]	0.089
Intake of vegetable protein, g/day	27.2 [21.3–32.9]	29.0 [23.4–35.7]	24.9 [20.1–30.5]	<0.001
Intake of vegetable protein, g/IBW/day	0.5 [0.4–0.6]	0.5 [0.4–0.6]	0.5 [0.4–0.6]	0.777
Animal/vegetable protein intake ratio	1.5 [1.1–2.0]	1.5 [1.1–1.9]	1.6 [1.2–2.1]	0.141
Intake of total fat, g/day	51.5 [40.4–67.9]	55.8 [44.1–70.2]	47.6 [38.2–60.6]	<0.005
Intake of fat intake, g/IBW/day	0.9 [0.7–1.2]	0.9 [0.7–1.1]	0.9 [0.7–1.2]	0.817
Intake of fat per energy intake, %	28.8 (6.5)	27.8 (6.6)	29.9 (6.3)	0.008
Intake of total carbohydrate, g/day	214.5 [162.2–256.0]	231.5 [184.8–275.1]	188.9 [146.1–229.2]	<0.001
Intake of carbohydrate, g/IBW/day	3.7 [2.9–4.5]	3.8 [3.1–4.6]	3.6 [2.7–4.4]	0.279
Intake of carbohydrate per energy intake, %	50.7 (8.9)	50.4 (9.2)	51.0 (8.5)	0.557
Intake of dietary fiber, g/day	11.2 [8.8–15.2]	11.1 [8.7–15.6]	11.4 [8.8–15.1]	0.772
Carbohydrate/fiber ratio	18.6 [14.3–24.0]	20.4 [15.9–25.4]	16.4 [14.0–20.5]	<0.001
Alcohol consumption, g/day	0.0 [0.0–3.6]	1.7 [0.0–16.7]	0.0 [0.0–0.0]	<0.001
Habitual miso consumption (−/+)	42/248	19/141	23/107	0.218
Miso soup intake, g/day	108.0 [55.4–1138.6]	110.9 [62.4–138.6]	96.0 [48.0–120.0]	<0.001

Number (absence [−]/presence [+]), mean (standard deviation) or median [interquartile range] was used for expression of data. Chi-square test, Student’s t test, or Mann-Whitney U test was used for evaluation of the difference between group. IBW, ideal body weight.

**Table 3 nutrients-13-01488-t003:** Clinical characteristics according to habitual miso consumption.

	Male *n* = 160			Female *n* = 130		
	Habitual Miso Consumption (−) *n* = 19	Habitual Miso Consumption (+) *n* = 141	*p*	Habitual Miso Consumption (−) *n* = 23	Habitual Miso Consumption (+) *n* = 107	*p*
Age, years	68.0 [64.0–79.5]	68.0 [63.0–74.0]	0.419	67.0 [61.0–73.0]	68.0 [62.5–73.5]	0.548
Diseases duration of diabetes, years	16.0 [12.5–27.0]	13.0 [8.0–21.0]	0.087	17.0 [8.0–22.0]	12.0 [4.5–18.0]	0.035
Family history of diabetes (−/+)	11/8	81/60	1.000	9/14	54/53	0.449
Height, cm	165.4 (6.8)	167.4 (6.5)	0.206	152.7 (5.1)	153.5 (5.5)	0.530
Body weight, kg	68.0 [57.7–73.3]	65.0 [59.8–72.0]	0.941	58.0 [54.6–66.0]	55.1 [49.6–60.6]	0.046
Body mass index, kg/m^2^	24.2 (2.8)	23.7 (3.3)	0.529	24.8 [23.2–29.0]	23.4 [21.1–26.1]	0.052
Systolic BP, mmHg	128.0 [114.0–137.0]	133.0 [122.0–144.0]	0.148	135.0 [132.0–143.0]	131.0 [119.0–143.0]	0.113
Diastolic BP, mmHg	75.5 (11.5)	79.9 (11.0)	0.099	74.0 [68.5–78.0]	76.0 [71.0–84.0]	0.224
Antihypertensive drugs (−/+)	5/14	61/80	0.246	7/16	50/57	0.231
Presence of hypertension (−/+)	4/15	50/91	0.323	2/21	38/69	0.023
Insulin (−/+)	17/2	101/40	0.167	13/10	80/27	0.132
Habit of smoking (−/+)	15/4	111/30	1.000	20/3	100/7	0.528
Habit of exercise (−/+)	11/8	65/76	0.470	11/12	53/54	1.000
Average HbA1c, mmol/mol	55.0 [46.2–60.6]	53.4 [49.0–58.3]	0.947	56.7 [50.4–68.7]	52.5 [46.9–58.1]	0.009
Average HbA1c, %	7.2 [6.4–7.7]	7.0 [6.6–7.5]	0.947	7.3 [6.8–8.4]	7.0 [6.4–7.5]	0.009
SD of HbA1c	0.27 [0.14–0.38]	0.20 [0.14–0.32]	0.603	0.37 [0.20–0.72]	0.21 [0.12–0.32]	0.004
CV of HbA1c	0.04 [0.02–0.05]	0.03 [0.02–0.05]	0.572	0.05 [0.03–0.09]	0.03 [0.02–0.04]	0.005
Fasting blood sugar, mmol/L	7.7 [6.2–9.1]	7.9 [6.9–10.1]	0.256	8.4 [7.0–10.6]	7.3 [6.2–8.5]	0.034
Creatinine, umol/L	82.2 [67.6–87.5]	73.4 [63.6–88.4]	0.296	55.7 [49.9–69.8]	55.7 [49.5–63.2]	0.784
eGFR, mL/min/1.73 m^2^	67.8 (19.5)	71.4 (20.4)	0.470	69.1 (24.6)	71.7 (16.4)	0.532
Uric acid, mmol/L	327.1 [291.5–348.0]	315.2 [267.7–368.8]	0.710	285.0 (52.0)	287.0 (63.9)	0.887
Triglycerides, mmol/L	1.1 [0.8–1.8]	1.3 [0.9–1.9]	0.718	1.3 [0.9–1.8]	1.2 [0.9–1.6]	0.481
HDL cholesterol, mmol/L	1.4 [1.3–1.5]	1.4 [1.2–1.7]	0.677	1.5 [1.4–1.7]	1.6 [1.3–1.8]	0.850
Triglycerides/HDL cholesterol, mmol/mol	0.8 [0.6–1.4]	0.8 [0.5–1.6]	0.924	0.9 [0.6–1.2]	0.7 [0.5–1.2]	0.426

Number (absence [−]/presence [+]), mean (standard deviation) or median [interquartile range] was used for expression of data. Chi-square test, Student’s t test, or Mann-Whitney U test was used for evaluation of the difference between group. BP, blood pressure; CV, coefficient of variation; eGFR, estimated glomerular filtration rate; HDL, high-density lipoprotein; and SD, standard deviation.

**Table 4 nutrients-13-01488-t004:** Habitual diet intake according to habitual miso consumption.

	Male *n* = 160			Female *n* = 130		
	Habitual Miso Consumption (−) *n* = 19	Habitual Miso Consumption (+) *n* = 141	*p*	Habitual Miso Consumption (−) *n* = 23	Habitual Miso Consumption (+) *n* = 107	*p*
Intake of total energy, kcal/day	1616.3 [1390.6–2014.2]	1832.2 [1588.0–2192.8]	0.148	1245.5 [930.3–1455.9]	1574.0 [1207.1–1811.7]	<0.001
Energy intake, kcal/IBW/day	27.3 [22.2–33.1]	30.3 [25.5–36.3]	0.311	24.9 [18.8–27.7]	29.4 [23.2–36.1]	<0.001
Intake of total protein, g/day	66.8 [53.3–83.3]	71.7 [57.4–88.2]	0.516	47.3 [35.7–64.0]	66.0 [54.9–84.2]	<0.001
Intake of protein, g/IBW/day	1.1 [0.9–1.4]	1.2 [0.9–1.4]	0.558	0.9 [0.7–1.2]	1.3 [1.0–1.7]	<0.001
Intake of protein per energy, %	15.3 [14.4–18.5]	15.5 [13.5–17.7]	0.513	17.0 (4.0)	18.1 (3.4)	0.189
Intake of animal protein, g/day	38.8 [32.6–51.2]	41.5 [31.5–56.9]	0.685	32.8 [19.9–41.4]	40.8 [30.3–54.1]	0.011
Intake of animal protein, g/IBW/day	0.6 [0.6–0.9]	0.7 [0.5–0.9]	0.833	0.6 [0.4–0.8]	0.8 [0.6–1.0]	0.010
Intake of vegetable protein, g/day	27.8 (10.2)	30.0 (8.8)	0.323	18.2 [13.7–22.0]	26.6 [21.3–30.9]	<0.001
Intake of vegetable protein, g/IBW/day	0.5 (0.2)	0.5 (0.1)	0.500	0.4 [0.3–0.4]	0.5 [0.4–0.6]	<0.001
Animal/vegetable protein intake ratio	1.4 [1.1–2.1]	1.5 [1.1–1.9]	0.704	1.6 [1.1–2.2]	1.6 [1.2–2.1]	0.860
Intake of total fat, g/day	65.9 [42.4–73.1]	55.2 [44.5–69.4]	0.299	42.4 [31.2–49.7]	48.6 [39.5–63.8]	0.020
Intake of fat intake, g/IBW/day	1.0 [0.8–1.2]	0.9 [0.7–1.1]	0.213	0.8 [0.6–0.9]	0.9 [0.7–1.3]	0.031
Intake of fat per energy intake, %	32.8 [27.0–35.8]	27.1 [24.0–30.5]	0.003	30.8 (7.4)	29.7 (6.1)	0.456
Intake of total carbohydrate, g/day	203.5 [169.9–238.0]	234.6 [186.2–280.0]	0.059	154.3 [122.6–190.1]	196.1 [151.9–238.0]	0.001
Intake of carbohydrate, g/IBW/day	3.4 [2.8–4.0]	3.8 [3.1–4.7]	0.121	3.1 [2.4–3.6]	3.7 [2.9–4.7]	0.002
Intake of carbohydrate per energy intake, %	49.3 (9.5)	50.5 (9.2)	0.586	50.9 (9.9)	51.0 (8.2)	0.929
Intake of dietary fiber intake, g/day	10.3 [7.6–13.5]	11.4 [8.8–15.8]	0.189	8.0 [5.5–9.4]	12.1 [9.2–15.3]	<0.001
Alcohol consumption, g/day	0.0 [0.0–0.6]	2.8 [0.0–17.6]	0.005	0.0 [0.0–0.0]	0.0 [0.0–0.0]	0.973

Number (absence [−]/presence [+]), mean (standard deviation) or median [interquartile range] was used for expression of data. Chi-square test, Student’s t test, or Mann-Whitney U test was used for evaluation of the difference between group. IBW, ideal body weight.

**Table 5 nutrients-13-01488-t005:** Multiple regression analysis on average HbA1c, SD of HbA1c and CV of HbA1c.

Male	Average HbA1c		SD of HbA1c		CV of HbA1c	
	β	*p*	β	*p*	β	*p*
Habitual miso consumption *	−0.086	0.290	0.017	0.817	0.028	0.721
Age, years	−0.201	0.025	−0.116	0.146	−0.113	0.140
Body mass index, kg/m^2^	0.066	0.442	0.072	0.337	0.067	0.408
Exercise †	0.099	0.232	−0.061	0.397	−0.065	0.412
Smoking ‡	−0.006	0.940	0.0004	0.996	−0.028	0.716
Insulin treatment §	0.139	0.101	0.018	0.805	0.025	0.755
Duration of diabetes, years	0.147	0.121	−0.052	0.529	−0.059	0.511
Average of HbA1c, %	-	-	0.496	<0.001	0.347	<0.001
Intake of energy, kcal/kg IBW/day	−0.061	0.512	0.168	0.042	0.172	0.054
Intake of carbohydrate per energy intake, %	−0.063	0.431	−0.061	0.382	−0.049	0.520
Intake of dietary fiber, g/day	−0.038	0.685	−0.082	0.317	−0.081	0.358
**Female**	**Average HbA1c**		**SD of HbA1c**		**CV of HbA1c**	
	**β**	***p***	**β**	***p***	**β**	***p***
Habitual miso consumption *	−0.251	0.009	−0.175	0.016	−0.185	0.022
Age, years	−0.126	0.193	−0.214	0.003	−0.198	0.014
Body mass index, kg/m^2^	0.059	0.533	0.073	0.297	0.057	0.464
Exercise †	−0.163	0.056	−0.128	0.045	−0.137	0.052
Smoking ‡	0.138	0.108	−0.089	0.161	−0.082	0.244
Insulin treatment §	0.048	0.586	0.064	0.327	0.091	0.214
Duration of diabetes, years	0.192	0.044	−0.098	0.169	−0.097	0.217
Average of HbA1c, %	-	-	0.572	<0.001	0.494	<0.001
Intake of energy, kcal/kg IBW/day	−0.007	0.948	0.011	0.886	0.012	0.889
Intake of carbohydrate per energy intake, %	0.028	0.738	0.036	0.569	0.013	0.847
Intake of dietary fiber, g/day	0.094	0.399	0.014	0.865	0.034	0.711

SD, standard deviation; CV, coefficient of variation; IBW, ideal body weight. * Habitual miso consumption was defined as non-habitual miso consumption (=0) or habitual miso consumption (=1). † Exercise was defined as non-regular exerciser (=0) or regular exerciser (=1). ‡ Smoking status was defined as non-smoker (=0) or smoker (=1). § Insulin sensitizers, insulin secretagogues, insulin treatment and nutrient load reducers were defined as without (=0) or with (=1).

## Data Availability

The datasets generated during and/or analysed during the current study are available from the corresponding author on reasonable request.

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
