# Peer review of "Habitual Miso (Fermented Soybean Paste) Consumption Is Associated with Glycemic Variability in Patients with Type 2 Diabetes: A Cross-Sectional Study"

_nutrients, 2021, doi:10.3390/nu13051488_

Round 1

Reviewer 1 Report

Takahashi, et al. found the relationship between miso consumption and glycemic variability among female patients with type 2 diabetes mellitus. 

Previous study showed the relationship between miso consumption between insulin resistance. In this cohort,  authors should provide the data regarding this issue. 

The authors divided two group: no-miso consumption and miso miso consumption. The authors shoud show dose-response relationshiop between consumption and glycemic variability. 

There must be several types of miso in Japan. Doesn't the difference in the type of miso affect the results? What is the most common type of miso in the region of this study?

How does the consumption of miso in this cohort compare to that of the general Japanese population?

In general, are there any characteristics of patients who consume a lot of miso soup? That characteristic could be a confounding factor.

Author Response

Response to Reviewer 1

Point 1

Takahashi, et al. found the relationship between miso consumption and glycemic variability among female patients with type 2 diabetes mellitus.

Previous study showed the relationship between miso consumption between insulin resistance. In this cohort, authors should provide the data regarding this issue.

Response

Thank you for your comment. As you say, to check the relationship between miso consumption and insulin resistance is important. Unfortunately, however, we did not measure insulin concentrations. Thus, we have checked triglycerides to HDL cholesterol ratio, which is reported as a marker of insulin resistance. Triglycerides to HDL cholesterol ratio was not different between people with and without habitual miso consumption (0.8 [0.5-1.6] vs. 0.8 [0.6-1.4], p = 0.924 in male 0.7 [0.5-1.2] vs. 0.9 [0.6-1.2], p = 0.426 in female), which might be because the effect of taking medication for dyslipidemia. According to your comment, we have mentioned this point as one of the limitations of this study in the Discussion section described as below.

Discussion (Line 303-314)

“Second, we did not assess the association between habitual miso consumption and insulin concentrations and postprandial blood sugar level in this study. We evaluated triglycerides to HDL cholesterol ratio, which is reported as a marker of insulin resistance [15], and found that there was no association between habitual miso consumption and triglycerides to HDL cholesterol ratio in this study (0.8 [0.5-1.6] vs. 0.8 [0.6-1.4], p = 0.924 in male 0.7 [0.5-1.2] vs. 0.9 [0.6-1.2], p = 0.426 in female). This might be because the effect of taking medication for dyslipidemia. Thus, it was not unclear that female with habitual miso consumption had lower insulin resistance than those without, and the association between habitual miso consumption and postprandial blood sugar level.”

  1. Fukuda Y, Hashimoto Y, Hamaguchi M, Fukuda T, Nakamura N, et al. Triglycerides to high-density lipoprotein cholesterol ratio is an independent predictor of incident fatty liver; a population-based cohort study. Liver Int 2016;36:713-720. https://doi.org/10.1111/liv.12977

Point 2

The authors divided two group: no-miso consumption and miso consumption. The authors should show dose-response relationship between consumption and glycemic variability.

Response

Thank you for your valuable suggestion. According to your suggestion, we have additionally investigated the effect of the miso soup intake, as a continuous variable, on HbA1c variability. In female, the miso soup intake was correlated with average (r = -0.290, p <0.001), SD (r = -0.316, p <0.001) and CV (r = -0.289, p <0.001) of HbA1c. On the other hand, the miso soup intake was not correlated with average (r = -0.079, p = 0.322), SD (r = -0.034, p = 0.667) and CV (r = -0.013, p = 0.874) of HbA1c in male. Thus, there is a relationship between miso soup consumption and glycemic variability in female but not in male. We have added these points in the Materials and Methods and the Results sections as below.

Materials and Methods (Line 137-141)

“Evaluation of correlation was assessed by Pearson’s correlation coefficient. Because the miso soup intake was a skewed variable, logarithmic transformation was done, before performing the investigation of the relationship of glycemic variability and log (miso soup intake +1).”

Results (Line 204-210)

“The correlation between the log (miso soup intake + 1) and average, SD, or CV of HbA1c were investigated. In female, the log (miso soup intake + 1) was correlated with average (r = -0.290, p <0.001), SD (r = -0.316, p <0.001) and CV (r = -0.289, p <0.001) of HbA1c. On the other hand, the log (miso soup intake + 1) was not correlated with average (r = -0.079, p = 0.322), SD (r = -0.034, p = 0.667) and CV (r = -0.013, p = 0.874) of HbA1c in male.”

Point 3

There must be several types of miso in Japan. Doesn't the difference in the type of miso affect the results? What is the most common type of miso in the region of this study?

Response

Thank you for your valuable comment. As you say, there is a possibility that different types of miso would affect the results, since several types of miso, such as rice koji miso, barley koji miso, soybean koji miso, and blended miso, were used in Japan. Unfortunately, however, we did not investigate type of miso in this study. Thus, we have mentioned this point as one of the limitations of this study in the Discussion section described as below.

Discussion (Line 314-319)

“Third, there is a possibility that different types of miso would affect the results, since several types of miso, such as rice koji miso, barley koji miso, soybean koji miso, and blended miso, were used in Japan. Unfortunately, however, we did not investigate type of miso in this study. Therefore, further research is needed the association between each type of miso glycemic variability.”

Point 4

How does the consumption of miso in this cohort compare to that of the general Japanese population?

Response

Thank you for your comment. In the previous study (reference of 11) with general Japanese people, the proportion of consuming miso soup every day were 38.8 % in male and 31.9 % in female, and the proportion of consuming miso soup over 4-5 times per one week were 63.1 % in male and 60.4 % in female. Another previous study reported the proportion of habitual miso consumption was 44.4 %. On the other hand, the proportion of habitual miso consumption in this study were 88.1 % in male and 82.3 % in female, which was higher than those in a previous study. We have added these points in the Discussion section as below.

Discussion (Line 238-242)

“In this study, 88.1 % of male and 82.3 % of female were categorized as habitual miso consumption, which was higher than the general population as previously reported [11,24]. There is a possibility that people with T2DM might consume miso more usually than general people.”

  1. Ikeda K, Sato T, Nakayama T, Tanaka D, Nagashima K, Mano F, et al. Dietary habits associated with reduced insulin resistance: The Nagahama study. Diabetes Research and Clinical Practice 2018;141:26–34. https://doi.org/10.1016/j.diabres.2018.04.006.
  2. Ito K, Miyata K, Mohri M, Origuchi H, Yamamoto H. The Effects of the Habitual Consumption of Miso Soup on the Blood Pressure and Heart Rate of Japanese Adults: A Cross-sectional Study of a Health Examination. Intern Med 2017;56:23-29. https://doi.org/10.2169/internalmedicine.56.7538

Point 5

In general, are there any characteristics of patients who consume a lot of miso soup? That characteristic could be a confounding factor.

Response

Thank you for your valuable comment. As you say, to consider characteristics of people who consume a lot of miso soup is important. A recent study, which clarify the usefulness of fermented soy product, showed that people, who took fermented soy product, tended to have more fruits and vegetables. In this study; thus, we have added the dietary fiber intake as covariates in Table 5. According to your comment, we have added these points in the Materials and Methods section as below.

Materials and Methods (Line 142-147)

“To investigate the effect of habitual miso consumption on glycemic parameters, including average, SD, and CV of HbA1c, multiple regression analyses was used, adjusting for potential cofounders; age, BMI, duration of diabetes, average HbA1c, insulin treatment [19], exercise habit, smoking status, intake of energy [20], intake of carbohydrate, and intake of dietary fiber [21].”

  1. Sakai R, Hashimoto Y, Hamaguchi M, Ushigome E, Okamura T, et al. Living alone is associated with visit-to-visit HbA1c variability in men but not in female in people with type 2 diabetes: KAMOGAWA-DM cohort study. Endocr J 2020;67:419-426. https://doi.org/10.1507/endocrj.EJ19-0436
  2. Kang HM, Kim DJ. Total energy intake may be more associated with glycemic control compared to each proportion of macronutrients in the korean diabetic population. Diabetes Metab J. 2012;36:300-306. https://doi.org/10.4093/dmj.2012.36.4.300
  3. Katagiri R, Sawada N, Goto A, Yamaji T, Iwasaki M, et al. Association of soy and fermented soy product intake with total and cause specific mortality: prospective cohort study. BMJ 2020;368:m34. https://doi.org/10.1136/bmj.m34

Reviewer 2 Report

This study is interesting to determine the miso intake effect on HbA1c in Japanese. However, since it is not a randomized clinical trial, it should include bigger subject numbers, and the miso intake needs to be shown as quantitative values. Statistical analysis should be corrected.    
English needs to be corrected. Some incomplete sentences were shown, and some incorrect sentences.
 Fasting serum glucose and serum insulin concentrations need to be provided, and HOMA-IR can be calculated. 
BDHQ needs to be explained in detail in the method section since miso intake was estimated from BDHQ. At least the components of BDHQ and how to estimate miso intake. 
Regular miso consumption needs to be defined better, like more than 1 /day or 3-4 times/ week.  The dishes containing miso are counted as miso intake. 
In statistics, the sample size used in the study since 290 subjects may not be sufficient in a cross-sectional study.  In Table 2, no habitual miso consumption was 19 and 23 for men and women, respectively. Do the authors think the number of subjects was sufficient? We better calculated the sufficient sample size, and if it is not sufficient, it should be mentioned in the limitation section.
In Table 1 and Table 4, statistical analysis of continuous variables was conducted with Mann-Whitney U. Did you check for the distribution of the subjects? In Tables, categorical variables were shown as number ( ) but no explanation of ( ). 
 In table 4, protein intake in women seems too low compared to other groups, and please check the number. Fat intake needs to be checked. The fat energy percent was lower, but the fat intake amount was higher between the miso intake and non-miso intake groups in women.
In table 5, the adjustment was not explained. It needs to be defined which variables were used for covariates. Since the subjects were heterogeneous even included taking insulin treatments and different energy intake between miso and non-miso intake groups, adjustments need to be done in the multiple regression analysis.       

Author Response

Response to Reviewer 2

Point 1

This study is interesting to determine the miso intake effect on HbA1c in Japanese. However, since it is not a randomized clinical trial, it should include bigger subject numbers, and the miso intake needs to be shown as quantitative values. Statistical analysis should be corrected.   

Response

Thank you for your suggestion. As you say, this study was not a randomized clinical trial, and it might be desirable to include bigger subject numbers. We have mentioned this point as one of the limitations of this study in the Discussion section described as below.

Discussion (Line 319-322)

“Fourth, because this was a cross-sectional study and the sample size was relatively small, we need further research, such as the study with more participants and a randomized clinical trial.”

In addition, according to your suggestion, we have additionally investigated the effect of the miso soup intake, as a continuous variable, on HbA1c variability. In women, the miso soup intake was correlated with average (r = -0.290, p <0.001), SD (r = -0.316, p <0.001) and CV (r = -0.289, p <0.001) of HbA1c. However, the miso soup intake was not correlated with average (r = -0.079, p = 0.322), SD (r = -0.034, p = 0.667) and CV (r = -0.013, p = 0.874) of HbA1c in men. Thus, there is a relationship between miso soup consumption and glycemic variability in women but not in men. In addition, we have corrected statistical analysis. We have added these points in the Materials and Methods and the Results sections as below.

Materials and Methods (Line 137-141)

“Evaluation of correlation was assessed by Pearson’s correlation coefficient. Because the miso soup intake was a skewed variable, logarithmic transformation was done, before performing the investigation of the relationship of glycemic variability and log (miso soup intake +1).”

Materials and Methods (Line 128-137)

“Data are shown as mean (SD), median [interquartile range] when the variables were skewed. A p-value of <0.05 was considered to be statistically significant.

Participants were categorized into two groups based on habitual miso consumption. Because the characteristics and dietary intakes were a different between sex, the data were analyzed male and female, separately. Evaluation of normal distribution was performed by Shapiro-wilk normality test. Student's t-test or Mann-Whitney U test for continuous variables and chi-square test for categorical variables was used for evaluation of the differences.”

Results (Line 204-210)

“The correlation between the log (miso soup intake + 1) and average, SD, or CV of HbA1c were investigated. In female, the log (miso soup intake + 1) was correlated with average (r = -0.290, p <0.001), SD (r = -0.316, p <0.001) and CV (r = -0.289, p <0.001) of HbA1c. On the other hand, the log (miso soup intake + 1) was not correlated with average (r = -0.079, p = 0.322), SD (r = -0.034, p = 0.667) and CV (r = -0.013, p = 0.874) of HbA1c in male.”

Point 2

English needs to be corrected. Some incomplete sentences were shown, and some incorrect sentences.

Response

Thank you for your comment. According to your comments we have revised them.

Point 3

Fasting serum glucose and serum insulin concentrations need to be provided, and HOMA-IR can be calculated.

Response

Thank you for your comment. Women with habitual miso consumption was lower fasting serum glucose than those without (7.3 [6.2-8.5] vs. 8.4 [7.0-10.6] mmol/L, p = 0.034). On the other hand, fasting serum glucose in men between with and without habitual miso consumption was not different (7.9 [6.9-10.1] vs. 7.7 [6.2-9.1] mmol/L, p = 0.256). We have added these points in the Materials and Methods and the Results sections as below.

Materials and Methods (Line 95-98)

“Venous blood was gathered from people with overnight fasting, and the levels of fasting blood sugar, triglyceride, high-density lipoprotein (HDL) cholesterol, uric acid, and creatinine (Cr) were measured.”

Results (Line 185-186)

“Fasting blood sugar in female with habitual miso consumption was lower than in those without it (p = 0.034).”

On the other hand, to check the relationship between miso consumption between insulin resistance is important. Unfortunately, however, we did not assess insulin concentrations. Thus, we have checked triglycerides to HDL cholesterol ratio, which is reported as a marker of insulin resistance. Triglycerides to HDL cholesterol ratio was not different in both people with and without habitual miso consumption (0.8 [0.5-1.6] vs. 0.8 [0.6-1.4], p = 0.924 in men 0.7 [0.5-1.2] vs. 0.9 [0.6-1.2], p = 0.426 in women), which might be because the effect of taking medication for dyslipidemia. According to your comment, we have mentioned this point as one of the limitations of this study in the Materials and Methods and the Discussion sections described as below.

Materials and Methods (Line 98-99)

“Triglycerides to HDL cholesterol ratio was calculated as triglycerides (mmol/L) divided by HDL cholesterol (mmol/L) [15].”

Discussion (Line 303-314)

“Second, we did not assess the association between habitual miso consumption and insulin concentrations and postprandial blood sugar level in this study. We evaluated triglycerides to HDL cholesterol ratio, which is reported as a marker of insulin resistance [15], and found that there was no association between habitual miso consumption and triglycerides to HDL cholesterol ratio in this study (0.8 [0.5-1.6] vs. 0.8 [0.6-1.4], p = 0.924 in male 0.7 [0.5-1.2] vs. 0.9 [0.6-1.2], p = 0.426 in female). This might be because the effect of taking medication for dyslipidemia. Thus, it was not unclear that female with habitual miso consumption had lower insulin resistance than those without, and the association between habitual miso consumption and postprandial blood sugar level.”

  1. Fukuda Y, Hashimoto Y, Hamaguchi M, Fukuda T, Nakamura N, et al. Triglycerides to high-density lipoprotein cholesterol ratio is an independent predictor of incident fatty liver; a population-based cohort study. Liver Int 2016;36:713-720. https://doi.org/10.1111/liv.12977

Point 4

BDHQ needs to be explained in detail in the method section since miso intake was estimated from BDHQ. At least the components of BDHQ and how to estimate miso intake.

Regular miso consumption needs to be defined better, like more than 1 /day or 3-4 times/ week.  The dishes containing miso are counted as miso intake.

Response

Thank you for your suggestion. Using BDHQ, the data of frequency of miso soup intake and the data of miso intake, which is automatically calculated according to the algorithm, were obtained. Frequency of miso soup intake was asked how often they consume miso soup per day; none, less than 1 cup, 1 cup, 2 cups, 3 cups, 4 cups, 5 cups, 6 or 7 cups, or 8 cups or more per day. The BDHQ asked participants how many cups of miso soup they consumed per day but not per week. Thus, in this study, people without habitual miso intake were defined as those who did not consume miso soup at all in a day. According to your suggestion, we have added these points in the Materials and Methods section as below.

Materials and Methods (Line 122-126)

“Frequency of miso soup intake was asked how often they consume miso soup per day; none, less than 1 cup, 1 cup, 2 cups, 3 cups, 4 cups, 5 cups, 6 or 7 cups, or 8 cups or more per day. We defined people without habitual miso consumption if they did not consume miso soup at all in a day [18].”

Point 5

In statistics, the sample size used in the study since 290 subjects may not be sufficient in a cross-sectional study. In Table 2, no habitual miso consumption was 19 and 23 for men and women, respectively. Do the authors think the number of subjects was sufficient? We better calculated the sufficient sample size, and if it is not sufficient, it should be mentioned in the limitation section.

Response

Thank you for your comment. As you say, the sample size was relatively small. Thus, we have mentioned this point as one of the limitations of this study in the Discussion section described as below.

Discussion (Line 319-322)

“Fourth, because this was a cross-sectional study and the sample size was relatively small, we need further research, such as the study with more participants and a randomized clinical trial.”

Point 6

In Table 1 and Table 4, statistical analysis of continuous variables was conducted with Mann-Whitney U. Did you check for the distribution of the subjects? In Tables, categorical variables were shown as number ( ) but no explanation of ( ).

 In table 4, protein intake in women seems too low compared to other groups, and please check the number. Fat intake needs to be checked. The fat energy percent was lower, but the fat intake amount was higher between the miso intake and non-miso intake groups in women.

Response

Thank you for your variable comment. According to your comment, we have checked the distribution of the subjects and reanalyzed. In Tables, categorical variables were shown as number (absence [-]/ presence [+]).

In table 4, protein intake was 66.0 [54.9-84.2] g/day in women with habitual miso consumption, and 47.3 [35.7-64.0] g/day in women without habitual miso consumption. As you say, the fat energy percent was lower, but the fat intake amount was higher in women with habitual miso consumption than in those without. This is because total energy intake in women with habitual miso consumption was higher than in those without it.

Tables

“Number (absence [-]/ presence [+]), mean (standard deviation) or median [interquartile range] was used for expression of data.”

Point 7

In table 5, the adjustment was not explained. It needs to be defined which variables were used for covariates. Since the subjects were heterogeneous even included taking insulin treatments and different energy intake between miso and non-miso intake groups, adjustments need to be done in the multiple regression analysis.       

Response

Thank you for your comment. As you say, we should show the explanation of covariates, included insulin treatments and energy intake. We chose these covariates because these covariates were reported to be associated with HbA1c levels. Thus, we have reanalyzed and mentioned in the Materials and Methods section as below.

Materials and Methods (Line 142-147)

“To investigate the effect of habitual miso consumption on glycemic parameters, including average, SD, and CV of HbA1c, multiple regression analyses was used, adjusting for potential cofounders; age, BMI, duration of diabetes, average HbA1c, insulin treatment [19], exercise habit, smoking status, intake of energy [20], intake of carbohydrate, and intake of dietary fiber [21].”

  1. Sakai R, Hashimoto Y, Hamaguchi M, Ushigome E, Okamura T, et al. Living alone is associated with visit-to-visit HbA1c variability in men but not in women in people with type 2 diabetes: KAMOGAWA-DM cohort study. Endocr J 2020;67:419-426. https://doi.org/10.1507/endocrj.EJ19-0436
  2. Kang HM, Kim DJ. Total energy intake may be more associated with glycemic control compared to each proportion of macronutrients in the korean diabetic population. Diabetes Metab J. 2012;36:300-306. https://doi.org/10.4093/dmj.2012.36.4.300
  3. Katagiri R, Sawada N, Goto A, Yamaji T, Iwasaki M, et al. Association of soy and fermented soy product intake with total and cause specific mortality: prospective cohort study. BMJ 2020;368:m34. https://doi.org/10.1136/bmj.m34

Reviewer 3 Report

In this study, the authors investigated the relationship between habitual miso consumption and glycemic control in patients with T2DM through a cross-sectional study. They conclude that habitual miso consumption is associated with good glycemic control in women, but not in men.

The topic addressed is interesting. I do see the need for some clarifications, however, and I hope that you can share my arguments below.

Major comments:

Discussion

1) Line 258-259

If you considered that beta-conglycinin as one of beneficial food ingredients, please explain why there was no change in triglycerides (Table 3, women).

2) Line 279-283

This study suggests that habitual miso consumption reduced only HbA1c levels. It seems inadequate to conclude that consumption of miso soup is associated with “good” glycemic control. The author should add a fasting blood sugar level and an after-meal blood sugar level if they want to make their discussion convincing.

Minor comments:

Materials & Methods

1) Line 124-125

“Habitual miso consumption was defined as regular consumption of miso soup.”

→ How often is the regular consumption of miso? Every meal? This should be much better clarified.

Discussion

2) Line 251-256

The authors’ argument is that habitual miso consumption is associated with good glycemic control in women, not but in men. However, reference 24, 25 and 27 reported that miso and isoflavones were effective in male model. I think authors should refer to studies on female model.

3) Line 256-258

“Therefore, isoflavones in miso may have inhibited the accumulation of visceral fat and contributed to good glycemic control in women.”

→ Please show data suggesting inhibition of visceral fat accumulation.

I hope that my comment is very useful for the improvement of the article.

Author Response

Response to Reviewer 3

In this study, the authors investigated the relationship between habitual miso consumption and glycemic control in patients with T2DM through a cross-sectional study. They conclude that habitual miso consumption is associated with good glycemic control in women, but not in men.

The topic addressed is interesting. I do see the need for some clarifications, however, and I hope that you can share my arguments below.

Response

Thank you for your valuable comment.

Point 1

Major comments:

Discussion

1) Line 258-259

If you considered that beta-conglycinin as one of beneficial food ingredients, please explain why there was no change in triglycerides (Table 3, women).

Response

Thank you for your valuable comment. As you say, beta-conglycinin, which included in miso, is one of beneficial food ingredients and reported decrease triglycerides levels. In this study, there was no difference between triglycerides level of people with habitual miso consumption and that of people without. This might be because that this study was observational study; and thus, the physicians prescribed the medication for dyslipidemia, if the triglycerides level were high. According to your comment, we have added this point in the Discussion section described as below.

Discussion (Line 279-288)

“A previous study revealed that β-conglycinin, which included in miso, is a beneficial food ingredient for prevention of obesity [34]. In addition, β-conglycinin was reported to decrease triglycerides levels [35,36]. Therefore, patients with habitual miso consumption may prevent visceral fat accumulation and improve glycemic control. However, there was no difference between triglycerides level of people with habitual miso consumption and that of people without in this study. This might be because that this study was observational study; and thus, the physicians prescribed the medication for dyslipidemia, if the triglycerides level were high.”

  1. Kohno M, Hirotsuka M, Kito M, Matsuzawa Y. Decreases in serum triacylglycerol and visceral fat mediated by dietary soybean β-conglycinin. Journal of Atherosclerosis and Thrombosis 2006;13:247–55. https://doi.org/10.5551/jat.13.247.
  2. Aoyama T, Kohno M, Saito T, Fukui K, Takamatsu K, et al. Reduction by phytate-reduced soybean beta-conglycinin of plasma triglyceride level of young and adult rats. Biosci Biotechnol Biochem 2001;65:1071-1075. https://doi.org/10.1271/bbb.65.1071
  3. Tachibana N, Iwaoka Y, Hirotsuka M, Horio F, Kohno M. Beta-conglycinin lowers very-low-density lipoprotein-triglyceride levels by increasing adiponectin and insulin sensitivity in rats. Biosci Biotechnol Biochem 2010;74:1250-1255. https://doi.org/10.1271/bbb.100088

Point 2

2) Line 279-283

This study suggests that habitual miso consumption reduced only HbA1c levels. It seems inadequate to conclude that consumption of miso soup is associated with “good” glycemic control. The author should add a fasting blood sugar level and an after-meal blood sugar level if they want to make their discussion convincing.

Response

Thank you for your comment. As you say, to show both fasting blood sugar level and after-meal blood sugar level is desirable. Thus, we have added the data of fasting blood sugar level, and found that women with habitual miso consumption was lower fasting blood sugar than those without (7.3 [6.2-8.5] vs. 8.4 [7.0-10.6] mmol/L, p = 0.034), whereas there was no difference of fasting blood sugar level in men with and without habitual miso consumption (7.9 [6.9-10.1] vs. 7.7 [6.2-9.1] mmol/L, p = 0.256). On the other hand, we did not measure after-meal blood sugar level. However, HbA1c levels partially reflect after-meal blood sugar level, although measuring after-meal blood sugar level directly is desirable. According to your comment, we have added these points in the Materials and Methods, the Results and the Discussion sections as below.

Materials and Methods (Line 95-98)

“Venous blood was gathered from people with overnight fasting, and the levels of fasting blood sugar, triglyceride, high-density lipoprotein (HDL) cholesterol, uric acid, and creatinine (Cr) were measured.”

Results (Line 185-186)

“Fasting blood sugar in female with habitual miso consumption was lower than in those without it (p = 0.034).”

Discussion (Line 303-314)

“Second, we did not assess the association between habitual miso consumption and insulin concentrations and postprandial blood sugar level in this study. We evaluated triglycerides to HDL cholesterol ratio, which is reported as a marker of insulin resistance [15], and found that there was no association between habitual miso consumption and triglycerides to HDL cholesterol ratio in this study (0.8 [0.5-1.6] vs. 0.8 [0.6-1.4], p = 0.924 in male 0.7 [0.5-1.2] vs. 0.9 [0.6-1.2], p = 0.426 in female). This might be because the effect of taking medication for dyslipidemia. Thus, it was not unclear that female with habitual miso consumption had lower insulin resistance than those without, and the association between habitual miso consumption and postprandial blood sugar level.”

Discussion (Line 325-327)

“This study demonstrated the relationship between habitual miso consumption and glycemic variability in female, but not in male, with T2DM.”

  1. Fukuda Y, Hashimoto Y, Hamaguchi M, Fukuda T, Nakamura N, et al. Triglycerides to high-density lipoprotein cholesterol ratio is an independent predictor of incident fatty liver; a population-based cohort study. Liver Int 2016;36:713-720. https://doi.org/10.1111/liv.12977

Point 3

Minor comments:

Materials & Methods

1) Line 124-125

“Habitual miso consumption was defined as regular consumption of miso soup.”

→ How often is the regular consumption of miso? Every meal? This should be much better clarified.

Response

Thank you for your comment. Using BDHQ, the data of frequency of miso soup intake and the data of miso intake, which is automatically calculated according to the algorithm, were obtained. Frequency of miso soup intake was asked how often they consume miso soup per day; none, less than 1 cup, 1 cup, 2 cups, 3 cups, 4 cups, 5 cups, 6 or 7 cups, or 8 cups or more per day. In this study, people without habitual miso intake were defined as those who did not consume miso soup at all in a day. According to your suggestion, we have added these points in the Materials and Methods section as below.

Materials and Methods (Line 122-126)

“Frequency of miso soup intake was asked how often they consume miso soup per day; none, less than 1 cup, 1 cup, 2 cups, 3 cups, 4 cups, 5 cups, 6 or 7 cups, or 8 cups or more per day. We defined people without habitual miso consumption if they did not consume miso soup at all in a day [18].”

Point 4

Discussion

2) Line 251-256

The authors’ argument is that habitual miso consumption is associated with good glycemic control in women, not but in men. However, reference 24, 25 and 27 reported that miso and isoflavones were effective in male model. I think authors should refer to studies on female model.

Response

Thank you for your variable suggestion. According to your suggestion, we have added the studies on female model in the Discussion section as below.

Discussion (Line 273-275)

“Moreover, soybean proteins and isoflavones have been reported to reduce visceral fat accumulation in animal models [31-33].”

  1. Aoyama T, Fukui K, Takamatsu K, Hashimoto Y, Yamamoto T. Soy protein isolate and its hydrolysate reduce body fat of dietary obese rats and genetically obese mice (yellow KK). Nutrition 2000;16:349–54. https://doi.org/10.1016/S0899-9007(00)00230-6.
  2. Davis J, Higginbotham A, O’Connor T, Moustaid-Moussa N, Tebbe A, Kim YC, et al. Soy protein and isoflavones influence adiposity and development of metabolic syndrome in the obese male ZDF rat. Annals of Nutrition and Metabolism 2007;51:42–52. https://doi.org/10.1159/000100820.
  3. Wu J, Wang X, Chiba H, Higuchi M, Nakatani T, et al. Combined intervention of soy isoflavone and moderate exercise prevents body fat elevation and bone loss in ovariectomized mice. Metabolism 2004;53:942-948. https://doi.org/10.1016/j.metabol.2004.01.019

Point 5

3) Line 256-258

“Therefore, isoflavones in miso may have inhibited the accumulation of visceral fat and contributed to good glycemic control in women.”

→ Please show data suggesting inhibition of visceral fat accumulation.

Response

Thank you for your comment. Although we should show the data about inhibition of visceral fat accumulation, we did not measure visceral fat in this study. However, we revealed that habitual miso consumption was associated with inhibition of abdominal obesity in a previous study. Thus, we have added this relationship in the Discussion section described as below.

Discussion (Line 275-277)

“In addition, we have revealed the association between habitual miso consumption and the abdominal obesity in a previous study [18].”

  1. Takahashi F, Hashimoto Y, Kaji A, Sakai R, Kawate Y, Okamura T, et al. Habitual miso (Fermented soybean paste) consumption is associated with a low prevalence of sarcopenia in patients with type 2 diabetes: A cross-sectional study. Nutrients 2021;13:1–14. https://doi.org/10.3390/nu13010072.

Round 2

Reviewer 1 Report

This manuscript had been revised well.

Reviewer 2 Report

Authors revised the manuscript according to the reveiwers suggestion.

I pleased with the revision.

Reviewer 3 Report

The presented manuscript is revised adequately.

This is an interesting observation that female with habitual miso consumption may have better glycemic control.